# β-carotene in Obesity Research: Technical Considerations and Current Status of the Field

**DOI:** 10.3390/nu11040842

**Published:** 2019-04-13

**Authors:** Johana Coronel, Ivan Pinos, Jaume Amengual

**Affiliations:** 1Department of Food Sciences and Human Nutrition, University of Illinois Urbana Champaign, Urbana, IL 61801, USA; acoronel@illinois.edu; 2Division of Nutritional Sciences, University of Illinois Urbana Champaign, Urbana, IL 61801, USA; ivanp2@illinois.edu

**Keywords:** Vitamin A, adipocyte, β-carotene oxygenase 1

## Abstract

Over the past decades, obesity has become a rising health problem as the accessibility to high calorie, low nutritional value food has increased. Research shows that some bioactive components in fruits and vegetables, such as carotenoids, could contribute to the prevention and treatment of obesity. Some of these carotenoids are responsible for vitamin A production, a hormone-like vitamin with pleiotropic effects in mammals. Among these effects, vitamin A is a potent regulator of adipose tissue development, and is therefore important for obesity. This review focuses on the role of the provitamin A carotenoid β-carotene in human health, emphasizing the mechanisms by which this compound and its derivatives regulate adipocyte biology. It also discusses the physiological relevance of carotenoid accumulation, the implication of the carotenoid-cleaving enzymes, and the technical difficulties and considerations researchers must take when working with these bioactive molecules. Thanks to the broad spectrum of functions carotenoids have in modern nutrition and health, it is necessary to understand their benefits regarding to metabolic diseases such as obesity in order to evaluate their applicability to the medical and pharmaceutical fields.

## 1. Introduction

Metabolic diseases are a growing cause of morbidity and mortality, becoming a heavy economic burden for patients and healthcare systems worldwide. More than 2.1 billion people worldwide were overweight or obese in 2014, and current predictions estimate that obesity will affect almost half of the world’s adult population by 2030 [1]. According to the US Centers for Disease Control and Prevention, the prevalence of obesity in the US was 39.8% in 2015–2016 [2], and the medical costs associated with obesity were approximately $2 trillion in 2014 [1]. While obesity per se is the direct cause of only a few disorders, such as bone and joint-related disease [3], obese individuals are more susceptible to suffer metabolic alterations leading to type 2 diabetes, high blood pressure, and heart disease. Obesity is also associated with depression and certain cancers, overall affecting longevity and life quality [4,5].

From a simplified point of view, obesity is a consequence of disproportionate energy intake in combination with a reduction in energy expenditure, which leads to a positive energy balance in the organism [6,7]. The hallmark of obesity is the excessive accumulation of triglycerides in adipocytes, the main cellular component of the white adipose tissue [8]. Even though multifactorial aspects such as genetic or epigenetic predisposition can cause obesity [9], acquired behaviors such as eating habits and reduced physical activity are the main contributing factors responsible for the development of this disease [10,11]. For example, the ingestion of only 5% more calories than those expended could result in the accumulation of approximately 5 kg of adipose tissue in just one year [12]. An adequate dietary intervention is the foundation of weight loss therapy, as it is easier for most obese people to achieve a negative energy balance by decreasing food intake than just by increasing physical activity [13,14,15]. Therefore, healthy eating is a great strategy for reducing obesity, and the key signature of a healthy diet is the high consumption of a plant-based diet [16]. 

While fruits and vegetables promote weight loss thanks to their macronutrient composition (e.g., elevated fiber, water, and complex carbohydrates), we cannot dismiss their content in micronutrients with bioactive properties, some of which have documented effects on energy metabolism [17,18]. Among these micronutrients, carotenoids appear as potential candidates to prevent and treat obesity [19]. Carotenoids are a diverse group of compounds responsible for most of the yellow, orange and red colors in fruits and vegetables. Over the years, researchers have attributed carotenoids multiple biological functions, and they are widely considered to be some of the most important bioactive compounds in our food [20]. Among the six most abundant carotenoids in plasma, β-carotene, α-carotene, and β-cryptoxanthin are provitamin A carotenoids. Two other carotenoids, lutein, and zeaxanthin, have an important role in vision as they largely accumulate in the human eye [21]. Lycopene, the last carotenoid on this list, plays a crucial role in preventing some types of cancer [22].

Several comprehensive reviews have recently been published discussing the effect of carotenoids in metabolic diseases [23,24], and so we will describe them briefly here. The focus of this review article is to combine the technical considerations and limitations on carotenoid research with a special emphasis on obesity research. For this, we will explore the mechanistic insights obtained from in vitro, cellular, and animal models, as well as observational and interventional studies in human subjects.

## 2. Vitamin A Sources in Mammals—Provitamin A Carotenoids 

Carotenoids are pigments synthesized mostly by photosynthetic organisms to function as light-harvesting scavengers during photosynthesis [25]. Chemically, carotenoids contain forty carbons, usually organized in a single tetraterpenoid chain with conjugated double bonds that are responsible for their coloration. Most carotenoids in our diet are cyclic on both ends, forming an ionone ring. Depending on the absence or presence of oxygen groups, carotenoids are classified as carotenes or xanthophylls, respectively. As such, β, β-carotene (β-carotene) (cyclic, two β ionone rings) and lycopene (acyclic) are two of the most common carotenes in nature, while lutein, zeaxanthin, and β-cryptoxanthin are the most abundant xanthophylls. There are approximately 650 carotenoids in nature, 50 are abundant in the human diet, and only 20 are significantly present in human plasma. Fruits and vegetables are the primary sources of dietary carotenoids, but some animal products such as eggs and salmon also contain these pigments in significant amounts [26]. 

Scientists have attributed many functions to carotenoids, most of which are positive on human health [20,27], although some notable exceptions have sparked intense debate in the scientific community [28,29,30,31]. Despite these controversies, carotenoids have an indisputable role in mammals, as some of them serve as a vitamin A precursor [32]. Only a handful of carotenoids have documented provitamin A activity, as they contain at least one unsubstituted β-ionone ring, required to produce vitamin A. Three of these carotenoids are carotenes: α,β-carotene (α-carotene), γ,β-carotene (γ-carotene) and β-carotene; and only one is a xanthophyll: β-cryptoxanthin. β-carotene, β-cryptoxanthin, and α-carotene are the most abundant provitamin A carotenoids in our diet (Table 1).

It is widely believed that β-carotene is the preferential source of vitamin A in mammals, since its two β-ionone rings can render two vitamin A molecules. This logical statement, however, has not been carefully characterized to date, as many other factors, such as the differential absorption and transport, cell type specificity, tissue storage or enzymatic accessibility, could define which provitamin A carotenoid is the preferred source for vitamin A production in humans. For example, some researchers postulate that β-cryptoxanthin could be a better source of vitamin A than β-carotene, presumably because of the presence of a hydroxyl group on its structure, which affects its solubility and absorption [34,35]. Previous studies have also shown that, unlike β-carotene, β-cryptoxanthin conversion to vitamin A involves a multi-step enzymatic process [36], introducing an extra layer of complexity that involves enzymatic activities and substrate partitioning. Furthermore, genetic variations localized in both the promoter and/or coding region transporters and enzymes involved in carotenoid metabolism are increasingly gaining attention, as their implications for human health could have a tremendous impact in the way we understand carotenoid metabolism [37,38,39,40].

Carotenoids are not the only source of vitamin A in animals. After carotenoids are ingested and converted to vitamin A, this fat-soluble vitamin is stored as retinyl esters in different organs, such as the liver and the adipose tissue. Therefore, the ingestion of certain organs such as the liver from other animals could be a significant source of vitamin A for humans (see Table 1). Vitamin A is also present in milk, providing this crucial nutrient to the newborn [41]. Since vitamin A deficiency is still a public health problem in more than half of the countries worldwide, fortification efforts, such as the addition of retinyl esters, are currently part of a preventive strategy carried out by many governments [42] 

Another strategy for reducing vitamin A deficiency consists of the genetic modification of certain crops poor in carotenoids, but highly consumed in countries exposed to vitamin A deficiency. A notable example of these strategies is the development of crops fortified with β-carotene such as golden rice [43] or cassava [44,45]. The biggest advantage of this second approach is that β-carotene is present in the food matrix and absorbed in the intestine by a protein-mediated process [46,47]. In the absence of enough vitamin A in the body, two intestinal proteins are dramatically upregulated to promote carotenoid uptake and conversion to vitamin A [48]. On the contrary, many subjects have reported vitamin A toxicity due to vitamin A ingestion with a wide variety of physiological manifestations such as liver damage, bone abnormalities, headaches, skin desquamation or death [49,50].

Taken together, the scientific consensus in the field considers provitamin A carotenoids as the preferred source of vitamin A for humans, and β-carotene as the main precursor of vitamin A [20,24,48].

## 3. β–Carotene Oxygenase 1 as Solely Responsible for Vitamin A Production in Mammals

β–carotene conversion to vitamin A was first postulated by Moore in 1930 after observing that vitamin A-deficient animals recovered health when fed β-carotene [51]. In 1965, two research groups showed that β-carotene was cleaved when incubated in the presence of rat intestine homogenates, and this cleavage occurred at the central 15-15’ double bond, yielding two molecules of vitamin A aldehyde (retinal). They proposed that the enzyme behaves like an oxygenase, naming it β-carotene 15-15’ dioxygenase (later named β-carotene oxygenase 1, BCO1), as it seemed to involve a dioxygenase reaction mechanism [52,53]. Recently, two laboratories confirmed these finding by using human recombinant BCO1 and its homolog in insects (NinaB) [54,55].

In 2000, von Lintig and Vogt cloned and identified for the first time an enzyme with β-carotene dioxygenase activity using *Drosophila melanogaster*, showing that this protein is a non-heme, iron-dependent soluble (cytosolic) enzyme [56]. In mammals, BCO1 is present in most tissues, but its expression is higher in the intestine and the liver indicating that β-carotene is largely converted to vitamin A in these shortly after its ingestion. However, since β-carotene is present in human plasma, and BCO1 has a broad expression pattern, it is plausible that peripheral tissues such as the adipose tissue can cleave β-carotene to form vitamin A locally [57]. This review discusses the implications of this pathway below. 

In 2001, von Lintig’s group cloned and characterized a second carotenoid cleaving enzyme named β-carotene oxygenase 2 (BCO2) [58]. These findings finally provided a mechanistic explanation by which asymmetric apocarotenoids are present in various animal tissues [59], a finding that, together with the cloning of BCO1, shook up the carotenoid field and facilitated many of the groundbreaking discoveries that would come after. 

## 4. β-Carotene and Obesity; Key Findings and Technical Limitations

In recent years, many research groups have studied the effects of carotenoids and apocarotenoids on energy metabolism. Due to its importance in nutrition and health, this field has been extensively reviewed in the past [23,24]. In this section, we will highlight the main findings related to β-carotene and its derivatives (retinoids) on energy metabolism, adipocyte function, and adipose tissue biology in various experimental models. We will also focus on some of the technical approaches and limitations of these studies, which we divided into cell culture, animal, and human studies. 

### 4.1. Cell Culture Studies—β-carotene and Adipogenesis

Cell culture studies represent a good strategy for elucidating the mechanism of action of bioactive compounds. These techniques allow the manipulation of genetic and environmental factors such as dose and time-dependent effects, unbiased screening of bioactive compounds, and tissue-specific targeted effects. However, cell culture studies using carotenoids are technically challenging due to their hydrophobicity, which often hampers their bioavailability in aqueous phases as cell culture media. Some carotenoids, such as β-carotene, are in relatively elevated concentrations in human plasma (Figure 1), but the direct dissolution of β-carotene into the media, for example at 2 µM using a common solvent such as dimethyl sulfoxide will result in its irreversible precipitation. Another strategy, the addition of strong detergents, such as Tween, is one of the most acceptable techniques used to promote the formation of β-carotene micelles [60,61,62]. This approach will lead to increased solubility in the media, but could cause toxicity and unexpected side effects, as these detergents produce holes in cell membranes. Less harmful approaches, such as cyclodextrin, serum, or purified lipoproteins, have been used in the past as an alternative [63], but in all cases, the stability of β-carotene must be considered, especially in the absence of antioxidants in solution [64].

Another hindrance to carotenoid research is the identification of the bioactive compound responsible for the cellular effects. As we mentioned above, most cell types express BCO1 and BCO2, which cleave carotenoids generating bioactive molecules with properties that often resemble those observed or described by the parent compound. For example, the exposure of either lycopene [65] or its BCO2-mediated cleavage product apo-10’-lycopenoic acid [66] attenuates the expression of pro-inflammatory cytokines in cultured adipocytes. Similarly to the transcriptionally active form of vitamin A retinoic acid, apo-10’-lycopenoic acid seems to activate the retinoic acid receptors (RARs) (see next section for more details), providing a mechanistic explanation of the effects observed in cell culture. Whether all BCO2- derived apocarotenoids can act as RAR ligands is not clear. Some studies show that some apocarotenoids do not bind RARs [67], while other studies suggest that some of them can act as RAR antagonists [68,69] (Figure 1). 

Altogether, the utilization of carotenoids in cell culture experiments can lead to misinterpretation of the results, as technical issues such as solubility and stability can arise. In addition, the presence and enzymatic activity of BCO1 and BCO2 will also affect the interpretation of the results. These issues must be addressed before interpreting the conclusions of an experiment using carotenoids in cell culture, a practice that often does not occur. 

### 4.2. Cell Culture Studies—Retinoic Acid and Adipogenesis 

The cleavage products of β-carotene by the action of BCO1, commonly named vitamin A, have a defined role in vision and gene expression, being involved in vital processes such as cellular differentiation, embryo development and immune system maturation [24,70,71,72,73]. Among the different forms of vitamin A, retinoic acid is the main regulator of gene expression by directly binding specific members of the nuclear receptor superfamily: RARs and the retinoid X receptors (RXRs). All-trans-retinoic acid, the predominant isomer in cells, has the highest binding affinity for RARs, while the 9-cis-retinoic acid isomer can bind both the RARs and the RXRs (See Figure 1). These nuclear receptors form different homo and heterodimer combinations (RXR:RAR, RXR:RXR, RXR:other), and modulate the transcription of over 650 genes. There are three different RAR and three RXR isoforms named alpha, beta and gamma (recently reviewed in [74]). Additionally, all-trans-retinoic acid, but not 9-cis-retinoic acid, can directly bind the PPARδ, an obligate RXR heterodimer [75]. Additionally, the endogenous retinoid 9-cis-13,14-dihydroretinoic acid (cis-DHRA) has been recently characterized as a novel RXR agonist [76]. While the precursor of this compound has not been established yet [77], its specificity for RXRs indicates that it could activate different pathways other than those activated by retinoic acid. 

The role of all-trans-retinoic acid (referred as retinoic acid from now on) in adipocyte biology was established many years ago through seminal work from Lazar’s group using cell culture models. They showed that retinoic acid inhibits adipocyte differentiation by blocking the expression of key adipogenic transcription factors [78]. Adding retinoic acid to fully differentiated adipocytes favors lipolysis instead, promoting adipocyte “browning” characterized by induction of UCP1 and brown/beige adipocyte markers [79,80]. 

β-carotene, the main provitamin A carotenoid, exerts similar effects to those observed by retinoic acid in mature brown adipocytes [81] and other cell types prone to accumulate lipids such as macrophages [62], indicating that β-carotene is efficiently cleaved by BCO1 to produce retinal, which is rapidly converted to its transcriptionally active form retinoic acid.

Due to the physiological relevance of white adipose tissue, and since this organ efficiently stores both β-carotene and vitamin A, observations by von Lintig’s group showed that β-carotene could also serve as a physiological source of retinoic acid in adipocytes. We observed that BCO1 expression is upregulated during adipocyte differentiation, which would facilitate the conversion of β-carotene to vitamin A in adipocytes as they become fully differentiated and capable of storing intracellular lipids. We also observed that exogenous β-carotene, but not retinol, triggered retinoic acid production in cultured adipocytes, indicating that β-carotene is the preferred source of retinoic acid in mature adipocytes over preformed vitamin A (retinol). Lastly, and in accordance with follow-up animal studies (see below), we observed that β-carotene-exposed adipocytes presented reduced lipolysis accompanied by a downregulation of PPARγ [82].

In summary, the use of retinoic acid to study the effects of β-carotene as a precursor of vitamin A will allow the researcher to bypass the difficulty of dissolving β-carotene in the media (β-carotene solubility in water is approximately 0.7 nM, while retinoic acid is 16 nM, values calculated using ALOGP software). Additionally, carotenoid oxygenases are not uniformly expressed in all cell types, and some carotenoids such as β-carotene can be cleaved by both BCO1 and BCO2, which could lead to confounding results. Lastly, while the concentration or retinoic acid to obtain significant effects in cell culture is generally considered pharmacological or even toxic, scientists should take into consideration that these compounds do not cross the lipid bilayer easily when added to the media [36]. 

### 4.3. Animal Models in Carotenoid Research

The use of animal models to resemble human physiology is a very common practice in biomedical research. Humans accumulate large amounts of carotenoids in tissues and plasma, but also cleave them to produce vitamin A and other apocarotenoids via BCO1 and BCO2. Similarly, some animal models, such as ferrets, gerbils and non-human primates, accumulate and cleave carotenoids. These models are expensive and the possibility to study certain diseases is very limited as not many genetically modified organisms are commercially available [83,84]. Rodents are the most common animal model in biomedical research, but do not accumulate carotenoids when fed at physiological doses. The cause of these interspecific differences could be explained by the catalytical properties of the carotenoid cleaving enzymes; while human or primate BCO1 and BCO2 show little enzyme activity in vitro, murine BCO1 and BCO2 are very active on cleaving carotenoids [85,86].

Based on these observations, scientists should be careful when choosing the adequate animal model to study carotenoid metabolism. For example, ferrets and gerbils are good animal models to study carotenoid absorption and biodistribution, while rodents are recommended if the goal is to examine the properties of the carotenoid cleavage products. Unfortunately, these recommendations are not always followed, and while some experts in the field have exposed multiple times the caveats of rodents to study intact carotenoids, many scientists choose to supplement carotenoids at supraphysiological concentrations, or even by injecting these dietary compounds intraperitoneally or intravenously without having a clear scientific justification.

It was not until over a decade ago when Johannes von Lintig and Adrian Wyss’ teams joined forces and developed two knockout mouse models lacking BCO1 and BCO2, respectively. In 2007, *Bco1*^−/−^ mice were characterized, and later, in 2011, *Bco2*^−/−^ mice followed. The most striking characteristic of both mouse models was their ability to accumulate carotenoids as it occurs in humans; *Bco1*^−/−^ accumulate β-carotene [87], while *Bco2*^−/−^ mice accumulate xanthophylls such as lutein, zeaxanthin [88], and β-cryptoxanthin [36], as well as the carotene lycopene [89]. While BCO1 is a cytosolic enzyme, BCO2 is present in the inner mitochondrial membrane where it protects this organelle against carotenoid accumulation [88,90,91]. Altogether, these experiments demonstrated that despite their similar catalytical properties, BCO1 and BCO2 have differential substrate specificity and subcellular localization, facilitating the compartmentalization of carotenoid metabolism. Taking β-carotene as an example, this carotenoid is cleaved to vitamin A in the cytosol (where BCO1 is present), but in case of an excess of intracellular β-carotene, BCO2 will prevent its accumulation in the mitochondria [88,90].

This is not the case for β-cryptoxanthin, another provitamin A carotenoid. While this carotenoid is the only xanthophyll with provitamin A activity, the exact mechanism of how vitamin A formation occurred was not clear. β-cryptoxanthin is one of the most abundant carotenoids in our diet (Table 1) and some authors consider this xanthophyll a better provitamin A carotenoid than β-carotene [34,35]. By using a combination of in vitro enzyme activity assays, cell culture experiments, and BCO1 and BCO2 deficient mice, we demonstrated that β-cryptoxanthin suffers a sequential cleavage to apo-10’-carotenal and retinal by BCO2 and BCO1, respectively [36]. These studies have recently been expanded, further confirming these findings [92] and pointing out the complexity of carotenoid metabolism and vitamin A production. 

Therefore, *Bco1*^−/−^ and *Bco2*^−/−^ mice seem to be a suitable model for studying carotenoid accumulation in humans, as they accumulate carotenoids as people do. However, these mouse models have a few limitations that researchers cannot forget. For example, as they are full-body knockouts, these mice do not cleave carotenoids at all. Additionally, feeding xanthophylls such as lutein or zeaxanthin to *Bco2*^−/−^ mice results in the accumulation of oxidized forms of these carotenoids, with little presence of the parent compound in tissues or sera [57,93]. This key difference should be considered if the research goal is to extrapolate the metabolism of xanthophylls to humans, in which the formation of oxidized carotenoids is minimal compared to rodents [57,94]. 

### 4.4. Animal Models—β-carotene and Obesity

Dietary interventions to study the effect of carotenoids in animal models generally describe their positive effects on the prevention and treatment of body weight gain, either with provitamin A carotenoids [95] or not [96,97,98,99]. While the effects of provitamin A carotenoids on obesity can be somehow explained by the conversion of these molecules to vitamin A (Figure 1), the mechanism by which the rest of carotenoids could reduce obesity remains largely elusive. Some authors hypothesize that these effects are mediated by apocarotenoid metabolites, while others suggest they are derived from the effects of the parent compound(s). Additionally, as mentioned before, some authors seem to ignore the importance of the carotenoid concentration in diet, which could potentially result in toxic side effects or alterations in food intake or nutrient absorption. 

In 2011, as part of a joint effort including eight laboratories, we studied the effect of β-carotene on obesity using wild-type and *Bco1*^−/−^ deficient mice. We observed that physiological amounts of dietary β-carotene reduced adipose tissue size in wild-type mice. These effects were associated with a global downregulation of adipogenic genes, most of which had a PPARγ-responsive element, similarly to our observations in cell culture [82]. On the contrary, *Bco1*^−/−^ mice did not show any significant changes upon β-carotene supplementation, despite *Bco1*^−/−^ mice accumulating large amounts of β-carotene in the adipose tissue. We also detected for the first time the accumulation of apo-10’-carotenol in *Bco1*^−/−^ mice, as the BCO2 asymmetric derivative of β-carotene. Yet, no significant changes were observed neither at the level of mRNA nor adipose tissue size or histological analyses. Overall, these results demonstrated for the first time that β-carotene reduces obesity only when it is converted to vitamin A, and that the accumulation of apo-10’-carotenol or β-carotene itself do not affect gene expression or obesity in vivo [57]. 

Since BCO2 can cleave β-carotene to produce apo-10’-carotenal, which accumulates as its reduced form apo-10’-carotenol, we next aimed to quantify the contribution of this enzyme in β-carotene metabolism. For this purpose, we used BCO1- and BCO2-deficient mice to generate BCO1/BCO2 double knockout mice. We observed that despite BCO2 can produce apo-10’-carotenoids, this pathway is marginal and BCO1 is the main β-carotene cleaving enzyme in mammals. Additionally, we described that apo-10’-carotenol can be stored, transported, and taken up by cells similarly to retinol. This observation is particularly important, as it shows that if BCO2 could easily convert β-carotene in cells, its cleavage product apo-10’-carotenol would interfere with vitamin A transporters/enzymes, potentially affecting key processes such as vision [36]. 

In summary, the use of animal models to study carotenoid physiology is a common practice that should be undertaken carefully. While some animal, such as ferrets and gerbils, are great models for mimicking carotenoid metabolism in humans, rodents can lead to confounding results, as they completely metabolize carotenoids when ingested. Currently, nearly 1,000 research articles have utilized rodents in carotenoid research, and only a few used BCO1 or BCO2-deficient mice to study carotenoid accumulation. The consideration of the adequate model to understand the function of these compounds, either as a whole or as an apocarotenoid metabolite(s), is crucial for the interpretation of the data. 

### 4.5. Animal Models—Vitamin A and Obesity

The effect of retinoic acid in obesity has been well documented over the years by different groups, and using different experimental methods. Most of these studies have focused on the treatment of adult mice, showing that retinoic acid exposure affects adipogenesis in animals [100], promotes adipose tissue browning [80,101], and favors an overall induction of fatty acid oxidation in adipocytes [102]. These observations were similar in other metabolic tissues, such as liver and muscle, both in cell culture [103,104] and animal models [105,106].

While the role of vitamin A in adipocyte differentiation was previously established in cell culture [78], and some reports implicate this compound in adult mice [100], we designed an experiment in which vitamin A was supplemented at physiological doses during adipose tissue development in vivo. To this end, we supplemented newborn rats with three times the normal content of retinyl esters in milk and determined if vitamin A exposure during adipogenesis could affect adipocyte differentiation and obesity later in life. This intervention was carried out during the entire lactation period (21 days) before weaning, which is the period when rats develop adipose tissue [107]. One set of rats was sacrificed after lactation, showing for the first time that vitamin A supplementation during adipogenesis promotes adipocyte proliferation in newborn animals, defined by an increased number of small adipocytes and an increased number of proliferating cell nuclear antigen (PCNA)-positive cells. A second set of rats were supplemented with retinyl esters following the same methodology, but subjected to a chow or high-fat diet after weaning time for 16 weeks. Two different cohorts of rats, with 5 animals/group and cohort, showed that rats receiving three times more retinyl esters during lactation and were fed high-fat diet, presented an increased obesity index and elevated leptin production in the white adipose tissue [108].

In our opinion, these findings clearly relate to those from Lazar’s group, where cultured preadipocytes exposed to retinoic acid showed reduced differentiation [78], which would translate to adipocyte proliferation. Our study showed that rats supplemented with retinyl esters as retinoic acid precursor during adipose tissue development increased adipocyte proliferation, confirming the role of retinoic acid and the importance of the exposure window. If too much vitamin A is administered during the proliferation stage of adipose tissue, this could lead to obesity later in life, while if retinoic acid is supplemented to mature adipocytes it will lead to weight loss [109]. 

### 4.6. Human Studies—β-Carotene and Obesity

It is common knowledge that diets rich in fruits and vegetables are associated with improved health and longevity and, since carotenoids are abundant in plants, many studies correlated the presence of these pigments with a healthy status in people (recently reviewed in [110,111,112]). Nevertheless, these observations could be merely associations based on the chemical properties of carotenoids, i.e., carotenoids are decreased in obesity because these compounds accumulate in the adipose tissue as this tissue works as a “trap” for lipophilic molecules. Osth’s work nicely supports this hypothesis by measuring β-carotene in the adipose tissue in lean and obese individuals. They showed that while obese people had lower β-carotene content in the adipose tissue when normalized to triglyceride tissue levels, these individuals had similar β-carotene content as lean control patients [113].

Many studies have tried to establish causation between elevated carotenoids in plasma and improved health status by using supplementation strategies, and while some studies provided positive results [114,115], the negative outcomes of two major clinical trials including β-carotene set back the possibility of β-carotene dietary carotenoid supplementation in humans. Both the alpha-tocopherol, β-carotene cancer prevention study (ATBC) and the β-carotene and retinol efficacy trial (CARET) showed that β-carotene supplementation promoted cancer progression in subjects exposed to asbestos and smoking [28,116]. Some authors have hypothesized that these adverse effects could be mediated by the anti-oxidant properties of β-carotene [117], supported by convincing data for some antioxidants such as vitamin E promoting cancer [118,119], although this nutrient was only present in the ATBC study, and not in the CARET study. Despite the efforts from many research groups, the relationship between cancer progression and β-carotene supplementation still remain elusive. It is possible that the elevated dose of β-carotene (20 and 30mg β-carotene/day) given during the ATBC and CARET studies to subjects already at risk of suffering cancer was the cause of these adverse effects [20]. Another possibility is that the purity of the β-carotene used in these trials was not adequate, as β-carotene is unstable when stored for a long period, or when it is exposed to elevated temperatures, light and/or oxygen ([120,121,122] and Figure 2). 

At the end of 2018, a group of researchers analyzed the health outcomes of over 29,000 participants of the ATBC study and correlated the baseline β-carotene plasma levels with overall and cause-specific mortality. Their study concluded that elevated β-carotene concentration is associated with lower cardiovascular disease, stroke, and cancer. This positive effect of β-carotene was dose-dependent, and the correction of their statistical models with the consumption of fruits and vegetables did not alter the outcome of this observation [123]. This study is particularly important, as it examines the effect of plasma β-carotene in a very large population [124], and agrees with a recent meta-analysis study showing that elevated β-carotene correlates with a lower incidence of metabolic syndrome [23]. Regarding obesity, β-carotene in plasma inversely correlate with body weight, as obese individuals have reduced levels compared to normal weight people (reviewed in [19]). Accordingly, Canas’ group showed that the supplementation of obese children with a carotenoid mixture showed a significant increase in plasma concentration of various carotenoids including β-carotene, accompanied by a decreased body mass index score, waist-to-height ratio, and subcutaneous adipose tissue content [125]. 

Nevertheless, β-carotene is the most abundant carotenoid in plasma, and the main precursor of vitamin A, that, as we previously reviewed, it is a crucial nutrient involved in the regulation of lipid metabolism in many organs, and is especially important in the adipose tissue, where it controls adipocyte differentiation, proliferation, and lipid metabolism. These observations are in accordance with the mounting evidence that points out that obesity [126] and other lipid-mediated disorders such as insulin resistance [23], fatty liver disease [117], or coronary artery disease [127] are associated with a lower retinoic acid status in human. Since β-carotene is the main precursor of vitamin A and, therefore, retinoic acid, the hypothesis that elevated β-carotene promotes retinoic acid signaling cannot be ignored. 

On the flip side of the coin, patients suffering from anorexia nervosa present elevated plasma carotenoid levels that can, in some cases, lead to hypercarotenemia and changes on skin color [128,129]. Scientists believed that a combination of an elevated intake of low-calorie foods rich on carotenoids such as carrots and spinach, a reduced metabolic conversion to vitamin A, and a reduced carotenoid deposition in adipose tissue could lead to hypercarotenemia [130,131]. In agreement with this hypothesis, patients suffering cachexia do not show signs of hypercarotenemia, although they present a reduced adipose tissue size, [132]. More research on this topic will be necessary to elucidate these interesting observations, as carotenoids could play a role on anorexia nervosa [132], or as carotenoids could worsen this disorder as some of them have leaning effects as they favor fatty acid oxidation via retinoic acid signaling. 

### 4.7. Human Studies—Synthetic Retinoids and Obesity

Considering the positive effects of retinoic acid on lipid metabolism in experimental models, and the indications that elevated retinoic acid status in human is associated with positive health outcomes, we would imagine that retinoic acid treatment in humans could reduce obesity and therefore work as an antiobesity agent. However retinoic acid is highly toxic for humans [133], and the use of less toxic, synthetic retinoids such as fenretinide could suppose a strategy to treat obese patients. Fenretinide is an anticancer drug currently employed in some clinical trials, alone (identifier NCT00546455) or in combination with other molecules [134]. The use of fenretinide as an antiobesity agent is supported by promising data in cell culture and animal studies describing its effects on preventing and treating obesity and some associated disorders [135,136,137,138]. We should be cautious, however, as some of the side effects of fenretinide and its metabolites include the reduction of vitamin A delivery to tissues, presumably through the binding of this molecule to the vitamin A transporter retinol-binding protein 4 [82,139,140,141,142]. 

Other synthetic retinoids, which are commonly used in the treatment of acne, have known effects on lipid metabolism. For example, oral treatment with Accutane (13- cis- retinoic acid) increases total triglyceride and LDL-cholesterol levels in some patients [143,144], but the effect on obesity has not been systematically studied. These effects could be mediated by alterations in hepatic lipid metabolism, as retinoic acid upregulates lipogenesis by inducing the expression of lipogenic genes [145,146], and its long-term exposure reduces lipolysis in certain cell types [147]. However, we observed that retinoic acid induces fatty acid oxidation in muscle cells [103], human hepatoma cells [104], and our recent data show that retinoic acid decreases triglyceride and cholesterol secretion in cultured hepatocytes (under preparation).

## 5. Conclusions

Mechanistic studies developed in cell culture and animal models show that β-carotene, the main precursor of vitamin A and retinoic acid, decreases obesity by promoting fatty acid oxidation in adipocytes and other tissues. Observational and interventional studies corroborate these findings and provide a mechanistic explanation by which the consumption of fruits and vegetables, rich in provitamin A carotenoids, prevent the development of obesity. 

In our opinion, the implementation of policies and strategies promoting the consumption of foods rich in provitamin A carotenoids could contribute to prevent the development of obesity and other metabolic diseases such as atherosclerosis and diabetes. Additionally, we also propose to introduce the quantification of retinoic acid and β-carotene into clinical practice as novel markers of health status. This is based on recent data obtained from clinical and preclinical models supporting the notion that low levels of these two compounds are tightly linked to the development of metabolic diseases. 

## Figures and Tables

**Figure 1 nutrients-11-00842-f001:**
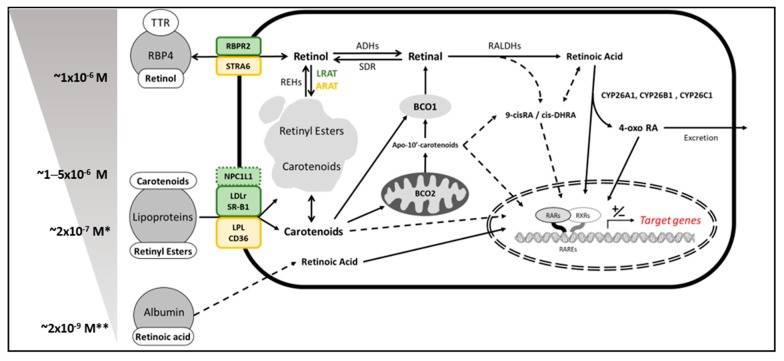
Schematic representation of carotenoid and vitamin A uptake and metabolism. **Left**: extracellular sources of provitamin A carotenoids (named carotenoids) and vitamin A and the relative concentration. * indicates that these sources can vary more than one order of magnitude depending on the fasting vs. fed conditions. Data show fasting values. **Right**: main proteins and conversion pathways involved in the uptake, cleavage/conversion and catabolism of carotenoids and vitamin A. For the purpose of this review, we only represented the proteins present in adipocytes and the hepatocytes in yellow and green, respectively. Dotted arrows are pathways not fully established. TTR, transthyretin; RBP4, retinol-binding protein 4; RBPR2, RBP4-receptor 2; STRA6, stimulated retinoic acid gene 6; NPC1L1, Niemann-Pick C1-Like 1; LDLr, Low-density lipoprotein receptor; SR-B1, scavenger receptor class B type 1; LPL, lipoprotein lipase; CD36, cluster of differentiation 36; REHs; Retinyl ester hydrolases; LRAT, lecithin:retinol acyl transferase, ARAT, acyl:retinol acyl transferase; ADHs, aldehyde hydrogenases; SDR, short-chain dehydrogenase/reductase; RALDH, retinaldehyde hydrogenases; 9-cisRA, 9-cis-retinoic acid; cis-DHRA, 9-cis-13,14-dihydroretinoic acid; CYP26s, cytochrome P450s; RARs, retinoic acid receptors; RXRs; retinoid X receptors; RAREs, retinoic acid-response element.

**Figure 2 nutrients-11-00842-f002:**
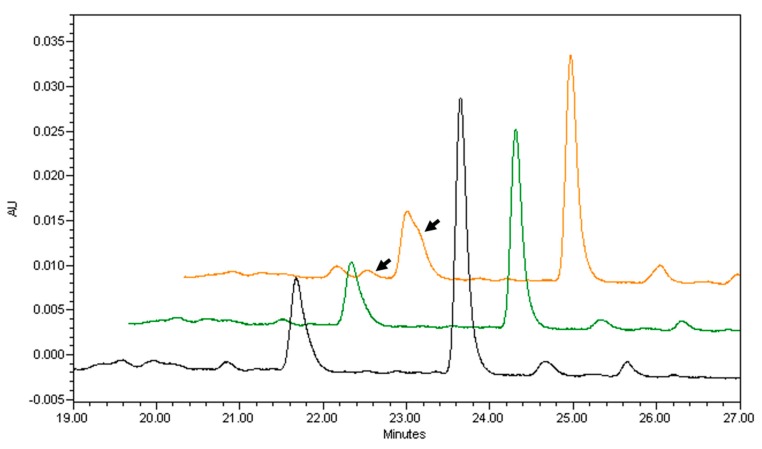
Purity and isomer variation between three commercial sources of β-carotene. HPLC chromatograms detected at 452 nm were obtained using YMC C30 column in three different commercially available β-carotene sources. The largest peak shows all-trans-β-carotene. Smaller peaks correspond to different carotenoid isomers (probably cis forms) in each commercial source. Arrows show the presence of two peaks only present in the orange chromatogram (probably β-carotene cis forms). AU, arbitrary units.

**Table 1 nutrients-11-00842-t001:** List of food sources abundant on pro-vitamin A carotenoids (red) and retinyl esters (blue). RAE, Retinol activity equivalent. Source; USDA Food composition database. Data expressed as μg/100 g food. Pro-vitamin A carotenoids in meat sources are very low or not present, while vegetables do not contain retinyl esters.

Source	β-Carotene	α-Carotene	β-Cryptoxanthin	Vitamin A, RAE
Peppers	42,891	6931	-	3863
Carrots	33,954	14,251	-	3423
Paprika	26,162	595	6186	2463
Pepper (spices)	21,840	-	6252	2081
Grape leaves	16,194	629	9	1376
Chili powder	15,000	2090	3490	1483
Sweet potatoes	12,498	47	-	1043
Pumpkin	6940	4795	-	778
Lettuce	5226	-	-	435
Squash	4226	834	3471	532
Seaweed	4872	-	-	406
Fish oil, cod liver	-	-	-	30,000
Liver (beef, other meats)	-	-	-	28,318
Lamb, liver	-	-	-	19,872
Duck, liver	-	-	-	11,984
Turkey, liver	-	-	-	10,751
Chicken, liver	-	-	-	4374

1 μg RAE = 1 μg retinol, 12 μg beta-carotene, 24 μg alpha-carotene, or 24 μg beta-cryptoxanthin. RAE conversion values obtained from [33].

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
