# Peer review of "β-carotene in Obesity Research: Technical Considerations and Current Status of the Field"

_nutrients, 2019, doi:10.3390/nu11040842_

Round 1
Reviewer 1 Report
The manuscript by Coronel et al, reviews the recent developments in understanding the role of provitamin A carotenoids and retinoids in obesity. The manuscript is based on studies reported in the last 5-8 years including many from the senior investigator. The authors have done a nice job of presenting both the current trends and controversies and also discussed the limitations of current approaches in the carotenoid/retinoid research field. Overall, this is a valuable contribution to the field and the manuscript is clear and generally well presented, but in some instances revisions would be advised to ensure the intended message is conveyed.
Table 1, just to clarify, is the absence of levels for provitamin A carotenoids in various animal sources (liver) an indication of absence or an indication of not having been measured?
Figure 1, it is not clear why the 9-cis-RA and cis-dihydro-RA forms are included if not discussed.
Please insert dash after trans or cis as in 9-cis-retinoic acid to follow standard notation
Line 222 "β-carotene is efficiently cleaved by BCO1 to produce retinoic acid" could be clarified that this is via aldehyde. Same with statements on line 309 and 314, apo-10’-carotenol is not a direct asymmetric cleavage product of β-carotene.
Line 241 while studies of the pKa of retinoic acid various environments are both old and few, in my interpretation, reference 83 seems to suggest that the pKA could be as high as 7-8 in micelles and bilayers suggesting that while retinoic acid may become deprotonated in neutral media (assuming a theoretical pK of 5) it readily crosses the membrane by flipping and from one leaflet to the other and by becoming protonated. Regardless, this is not especially clear.
Lines 363-68 reviewing the nice work by Osth et al and other studies which show that increased fat mass leads to decreased adipose concentration but not content of beta-carotene in obese state. It would be instructive to discuss if obesity also causes reduced bioavailability or reduced RA signaling in obese state. This is not clear from the discussion.
Line 379 the sentence starting with "the efforts" requires a verb accord and a preposition as in "Despite, the efforts from many research groups, in finding the relationship between cancer progression and β-carotene supplementation still remains elusive" or something similar. Please carefully proofread the manuscript for grammar and clarity.
Line 391 please specify that the largest peak shows "all-trans-β-carotene".
Line 412 replace revised with reviewed or similar.
Line 415, please revise "increasing number of evidence" to mounting evidence or other common phrasing.
Line 420, is the intended message of the paragraph that hypercarotenemia is a result of anorexia or that carotenoids could play a role on anorexia, or both. This could be simplified.
Author Response
Reviewer #1
The manuscript by Coronel et al, reviews the recent developments in understanding the role of provitamin A carotenoids and retinoids in obesity. The manuscript is based on studies reported in the last 5-8 years including many from the senior investigator. The authors have done a nice job of presenting both the current trends and controversies and also discussed the limitations of current approaches in the carotenoid/retinoid research field. Overall, this is a valuable contribution to the field and the manuscript is clear and generally well presented, but in some instances, revisions would be advised to ensure the intended message is conveyed.
Response: We appreciate the reviewer’s compliments regarding this manuscript, and its contribution to this research field. We also acknowledge the comments concerning structure and clarity and, as follows, we try to satisfy the proposed revisions.
Comments
1. Table 1, just to clarify, is the absence of levels for provitamin A carotenoids in various animal sources (liver) an indication of absence or an indication of not having been measured?
Response: Thank you for your comment. The reason why we did not add the amount of provitamin A carotenoids in the table is because, in most cases, the levels of these compounds were not measured. In other cases, the amounts were very little. To clarify this, we now indicate on the table’s legend that “Pro-vitamin A carotenoids in meat sources are very low or not present, while vegetables do not contain retinyl esters”.
As a follow up, we would like to mention that carotenoid metabolism and accumulation in poultry and other types of meat or fish is very complex. Some animals accumulate carotenoids, therefore they could be a source of these compounds in our diet. Interestingly, new evidences show that some animals such as beef, sheep, and chicken can accumulate xanthophylls due to the presence of genetic variants on the carotenoid-cleaving enzyme BCO2, for at least some of the cases [1-3].
We did not discuss this exciting topic in the current review, as it is out of the scope of our work.
2. Figure 1, it is not clear why the 9-cis-RA and cis-dihydro-RA forms are included if not discussed.
Response: We apologize for this mistake. While the role of 9-cis-RA was discussed in the text (Lines 207-209 of the current version), we did not include any information about cis-DHRA. We have incorporated some of the recent findings about this compound in the current version of the manuscript, as it follows:
“Additionally, the endogenous retinoid 9-cis-13,14-dihydroretinoic acid (cis-DHRA) has been recently characterized as a novel RXR agonist [77]. While the precursor of this compound has not been established yet [78], its specificity for RXRs indicates that it could activate different pathways other than those activated by retinoic acid.”
3. Please insert dash after trans or cis as in 9-cis-retinoic acid to follow standard notation
Response: Thanks for the recommendation, all nomenclatures have been modified in accordance with the standard notation. We left the tacking changes on the document, in case the reviewer wants to verify it.
4. Line 222 "β-carotene is efficiently cleaved by BCO1 to produce retinoic acid" could be clarified that this is via aldehyde. Same with statements on line 309 and 314, apo-10’-carotenol is not a direct asymmetric cleavage product of β-carotene.
Response: Thank you for indicating this, we apologize for skipping the intermediate step. We have corrected and clarified these statements by adding;
Line 222 (current version 227-228) – we added “indicating that β-carotene is efficiently cleaved by BCO1 to produce retinal, which is rapidly converted to its transcriptionally active form retinoic acid”
Line 309 (current version 314) – we replaced “cleavage product” by derivative
Line 314 (current version 318-319) – we clarified this statement adding the following sentence. Since BCO2 can cleave β-carotene to produce “apo-10’-carotenal, which accumulates as its reduced form apo-10’-carotenol”
5. Line 241 while studies of the pKa of retinoic acid various environments are both old and few, in my interpretation, reference 83 seems to suggest that the pKA could be as high as 7-8 in micelles and bilayers suggesting that while retinoic acid may become deprotonated in neutral media (assuming a theoretical pK of 5) it readily crosses the membrane by flipping and from one leaflet to the other and by becoming protonated. Regardless, this is not especially clear.
Response: We agree with your comment about Noy’s paper. To avoid misinterpretations, we have removed this statement and the corresponding reference.
6. Lines 363-68 reviewing the nice work by Osth et al and other studies which show that increased fat mass leads to decreased adipose concentration but not content of beta-carotene in obese state. It would be instructive to discuss if obesity also causes reduced bioavailability or reduced RA signaling in obese state. This is not clear from the discussion.
Response: Thank you for your comment. Indeed, it is not clear if obesity causes a reduced bioavailability of beta-carotene, therefore we did not discuss it in our review. However, multiple studies show that retinoic acid signaling in obesity, as well as in other metabolic diseases such as insulin resistance, fatty liver disease, and coronary artery disease is reduced in human and animal models. Accordingly, we discuss these findings in the same section (section 4.6.), a few paragraphs below (Lines 417-420).
7. Line 379 the sentence starting with "the efforts" requires a verb accord and a preposition as in "Despite, the efforts from many research groups, in finding the relationship between cancer progression and β-carotene supplementation still remains elusive" or something similar. Please carefully proofread the manuscript for grammar and clarity.
Response: Thanks for the correction and advice. We have rephrased this sentence.
8. Line 391 please specify that the largest peak shows "all-trans-β-carotene".
Response: We apologize for this. We added “all-trans” to the figure legend.
9. Line 412 replace revised with reviewed or similar.
Response: “Revised“ has been replaced by “reviewed”.
10. Line 415, please revise "increasing number of evidence" to mounting evidence or other common phrasing.
Response: We replaced “increasing number of evidence” by “mounting evidence”.
11. Line 420, is the intended message of the paragraph that hypercarotenemia is a result of anorexia or that carotenoids could play a role on anorexia, or both. This could be simplified.
Response: As the reviewer pointed out, this paragraph was not very clear. We have rephrased it to simplify it and make it more intelligible. Now the new paragraph is as follows:
“On the flip side of the coin, patients suffering from anorexia nervosa present elevated plasma carotenoid levels that can, in some cases, lead to hypercarotenemia and changes on skin color [129, 130]. Scientists believed that a combination of an elevated intake of low-calorie foods rich on carotenoids such as carrots and spinach, a reduced metabolic conversion to vitamin A, and a reduced carotenoid deposition in adipose tissue could lead to hypercarotenemia [131,132]. In agreement with this hypothesis, patients suffering cachexia do not show signs of hypercarotenemia, although they present a reduced adipose tissue size [133]. More research on this topic will be necessary to elucidate these interesting observations, as carotenoids could play a role on anorexia nervosa [133] or as carotenoids could worsen this disorder as some of them have leaning effects as they favor fatty acid oxidation via retinoic acid signaling.”
References
1. Manuchar, S., Omer, M., & Rahman, H. (2015). Differentiation between Normal and Abnormal Yellow Colour of Beef Carcasses in Sulaimani New Slaughterhouse. Nutr Food Technol, 1(1). https://doi.org/10.16966/2470-6086.105
2. Eriksson, J., Larson, G., Gunnarsson, U., Bed’hom, B., Tixier-Boichard, M., Strömstedt, L., … Andersson, L. (2008). Identification of the yellow skin gene reveals a hybrid origin of the domestic chicken. PLoS Genetics, 4(2), e1000010. https://doi.org/10.1371/journal.pgen.1000010
3. Våge, D. I., & Boman, I. A. (2010). A nonsense mutation in the beta- carotene oxygenase 2 (BCO2) gene is tightly associated with accumulation of carotenoids in adipose tissue in sheep (Ovis aries). BMC Genetics, 11, 10. https://doi.org/10.1186/1471-2156-11-10
Reviewer 2 Report
Review
β-carotene in Obesity Research: Technical Considerations and Current Status of the Field
Johana Coronel, Ivan Pinos, Jaume Amengual,
First, an interesting title and topic: β-carotene in obesity research.
However, from my point of view the review is too extensive, everything about vitamin A and carotenoids, and then the essential topic β-carotene in human health, to regulate adipocyte biology and its role in obesity and in human studies comes too short. Most facts are often described too imprecisely; often only about vitamin A or the ‘carotenoids’ as a whole instead of β-carotene is talked and written about.
After a long introduction, and several subitems, I always wondered where the authors wanted to go, and/or what they really want to achieve with this review. Most of the facts are known, and when it comes to concrete studies where a positive effect of vitamin A or β-carotene should be mentioned, it is very ‘spongy’ and several times incorrectly summarized and cited.
Finally, do the authors would like to recommend the higher consumption of fruits and vegetables, as the high intake of pro-vitamin A carotenoids seems to prevent the development of obesity, arteriosclerosis and diabetes – then please work out more detailed this, would be very interesting and important. Or do they want β-carotene and retinoic acid to be measured as a health status marker in future studies? or would the authors like to point out above all the weaknesses and lack of expressiveness of the previous studies in relation to β-carotene in the obesity research?
At the beginning the cell cultures studies were probably the most promising in terms of effects by β-carotene on reduced lipolysis. On the other hand, the poor transferability of the results (technical limitations) and the probably very high necessary and already toxic concentrations of retinoic acid (instead of β-carotene!) in the cell cultures experiments were pointed out. Concerning the animal studies, limitation regarding the β-carotene studies was also shown in particular. However, the authors own studies showed that too much retinyl ester in form of retinoic acid during adipose tissue development in rats could lead to obesity in later life [Ref 110]. Here, of course, I am missing a clear study regarding β-carotene, the actual topic of the review.
However, I would really advise the authors to start again from the beginning, and clearly work out the role of β-carotene in obesity research. To summarize and shorten the first parts in the introduction, cell- and animal studies with most relevant facts, in order to put the facts into relation with the relevant human studies on β-carotene and obesity research. I would especially work out the relevant human studies more thoroughly; especially here obvious mistakes in the citation had been also noticed.
After Line 88: Is table 1 really necessary??? If yes please provide conversion factors
In line 114: ‘carotenoid toxicity’ – I never heard about carotenoid toxicity; probably you mean vitamin A toxicity, and this is what you explain in the following sentences.
Comment, line 383-86: where is it proven that 9-cis or 13-cis beta-carotene have an adverse effect on human health? Are there studies (cell cultures, animals studies, human studies) showing effect by cis-isomers of beta-carotene? we also find cis isomer of beta-carotene in many food samples such as in fruits and green leafy vegetables.
Citations, references: I can't control all statements and literature cited; but the following are very general, imprecise and don't really reflect the details and results of the summarized studies.
Line108-110: ‘For examples, vitamin A fortification, of margarine in Denmark as well sugar fortification in Central America and Africa has considerably reduced the prevalence of vitamin A deficiency’[42-44].
First, what have it to do with beta-carotene? Did vitamin A fortification of margarine in Demark reduce vitamin A deficiency? I cannot find sugar fortification (with vitamin A?) in Central America (Brazil?) and Africa in the following 3 papers [42-44].
42. Keller, A., et al., A retrospective analysis of a societal experiment among the Danish population suggests that exposure to extra doses of vitamin A during fetal development may lower type 2 diabetes mellitus (T2DM) risk later in life. Br J Nutr, 2017. 117(5): p. 731-736.
Individuals from birth cohorts with the higher prenatal vitamin A exposure (born 1 December 1962–31 March 1964) and those with lower prenatal exposure (born 1 September 1959–31 December 1960) were followed up with regard to development of type 2 diabetes mellitus (T2DM) before 31 December 2012 in the Danish National Diabetes Registry and National Patient Register. Logistic regression revealed that higher fetal exposure (margarine vitamin A (RDA) concentration increased by 25%: retinol changed from 4.2 µg to 6 µg/g and b-carotene from 3.6 µg/g to 3 µg/g). Among the 193 803 included individuals, 101 178 and 92 625 were exposed and less-exposed, respectively, to the higher extra levels of vitamin A prenatally. The individuals exposed to higher vitamin A from fortification were less likely to develop T2DM (OR 0·88; 95 % CI 0·81, 0·95, P = 0·001) – But prevalence of vitamin A deficiency was not measured; the models only assessed risk of T2DM!
43. Palmer, A.C., et al., Provitamin A-biofortified maize increases serum beta-carotene, but not retinol, in marginally nourished children: a cluster-randomized trial in rural Zambia. Am J Clin Nutr, 2016. 104(1): p.181-90.
A study in Zambia assessed the impact of orange -maize meal (n=543 children) vs. white maize meal (n=481 children) among 4-8 years old children; 6-monthn biofortified maize intervention (higher daily intake of b-carotene – 133 µg daily RAE) revealed significant higher serum b-carotene (+0,14 µmol/L), but no difference in retinol concentrations between trial arms – no reduction in vitamin A deficiency and no sugar fortification!?!
44. Silva, L.L.S., et al., The impact of home fortification with multiple micronutrient powder on vitamin A status in young children: A multicenter pragmatic controlled trial in Brazil. Matern Child Nutr, 2017. 13(4).
A study in Brazil among 6-8 month old children, who received a multi-micronutrient powder (consisting of vitamin A, D, C, E, , B-vitamins (B1-B12) iron, zinc , copper, Iodine), who were followed for 4-6 month (n=399) and had significantly higher serum retinol, and therefore lower prevalence of vitamin A deficiency, but also lower prebalence of anemia and iron deficiency compared to 11-14 months old controls (n=395) - again, no sugar fortification!
Line 405-410
‘Accordingly, Canas’ group showed that the supplementation of obese children with a carotenoid mixture showed a positive correlation between β-carotene concentration and decreased body mass index score, waist-to-height ratio, and adipose tissue size [126]. Similarly, supplementation of β-cryptoxanthin, another provitamin A carotenoid, to obese Japanese women resulted in increased serum levels of β-cryptoxanthin, reduction in body weight, visceral fat and waist circumference [127, 128]’.
126. Canas, J.A., et al., Effects of Mixed Carotenoids on Adipokines and Abdominal Adiposity in Children: A Pilot Study. J Clin Endocrinol Metab, 2017. 102(6): p. 1983-1990.
A very small pilot study (n= 17 children, 10 years old) assessing mixed carotenoid supplementation (MCS) versus placebo on adipokines and the accrual of abdominal adiposity in children with simple obesity. After 6 month the children who received carotenoid mixture (n=8) had a significantly higher alpha- and beta-carotene, lutein and zeaxanthin, but significantly reduction in serum retinol; BMI z-scores increased by 4% in the placebo group but decreased by 4% in the MCS group; further high-molecular-weight adiponectin (HMW-ADI) increased by 79% in the MCS vs. placebo group. – the authors suggested a putative beneficial role of beta-carotene in the prevention and management of obesity.
127. Hirose, A., et al., Higher intake of cryptoxanthin is related to low body mass index and body fat in Japanese middle-aged women. Maturitas, 2017. 96: p. 89-94.
Different intake pattern regarding higher intake of cryptoxanthin (especially high in mandarins) was associated with low BMI (< 22 kg/m2 (n=56) vs >22 kg/m2 (n=23)) and low body fat percentage (<25%, n=35 vs >25%, n=53).
Limitation of the study: small number (n=88 women), data were taken from a baseline survey; most women were not obese, and plasma cryptoxanthin were not measured – this was not a supplementation study, and cryptoxanthin was not supplemented to obese women as written in line 408!
128. Iwamoto, M., et al., Supplementation of highly concentrated beta-cryptoxanthin in a satsuma mandarin beverage improves adipocytokine profiles in obese Japanese women. Lipids Health Dis, 2012. 11: p. 52.
An intervention study in which β-cryptoxanthin (4.7 mg/day) was given for 3 weeks to 17 moderately obese postmenopausal women. The result indicated no changes in body weight! serum cryptoxanthin increased by 4-fiold, and high molecular weight-adiponectin significantly. The authors concluded that increasing the intake of β-cryptoxanthin to approximately 4 mg per day for 3 weeks may have beneficial effects on the serum adipocytokine status and consequently alleviate progression of metabolic syndrome.
Supplementation of β-cryptoxanthin to moderate obese (BMI=27 kg/m2) Japanese women DID NOT result in reduction of body weight, visceral fat and waist circumference, as mentioned in Line 409-10!! Visceral fat and waist circumference were not measured at all!
I cannot control all the 149 citations, and I really hope that most citations reflect the truth better than the above-mentioned lines and references.

Author Response
Reviewer #2
First, an interesting title and topic: β-carotene in obesity research.
However, from my point of view the review is too extensive, everything about vitamin A and carotenoids, and then the essential topic β-carotene in human health, to regulate adipocyte biology and its role in obesity and in human studies comes too short. Most facts are often described too imprecisely; often only about vitamin A or the ‘carotenoids’ as a whole instead of β-carotene is talked and written about.
Response: Thank you for acknowledging the interest of the main topic of this review and for your criticism.
While the topic of the special issue we want to include our work in is “Carotenoids and Human Health”, we consider that mechanistic studies using cell culture and animal models are crucial to understand the findings observed in human subjects. Accordingly, we focused large part of our review on these type of studies.
After a long introduction, and several subitems, I always wondered where the authors wanted to go, and/or what they really want to achieve with this review. Most of the facts are known, and when it comes to concrete studies where a positive effect of vitamin A or β-carotene should be mentioned, it is very ‘spongy’ and several times incorrectly summarized and cited.
Response: We agree with the reviewer pointing out that the organization of the review contains several subitems, maybe too many. However, we believe these subitems help the reader to find or skip specific sections of the manuscript.
We also agree with the reviewer that most, if not all the facts explained in this review are known. The goal of this review was to provide an update on this important research field, but also indicate the technical limitations and cautionary notes that researchers should follow when working on carotenoid research.
Finally, do the authors would like to recommend the higher consumption of fruits and vegetables, as the high intake of pro-vitamin A carotenoids seems to prevent the development of obesity, arteriosclerosis and diabetes – then please work out more detailed this, would be very interesting and important. Or do they want β-carotene and retinoic acid to be measured as a health status marker in future studies? or would the authors like to point out above all the weaknesses and lack of expressiveness of the previous studies in relation to β-carotene in the obesity research?
Response: Following the reviewer’s suggestion, we rephrased our concluding remarks (section 5) to provide our final opinion and a “take-home message” and some future directions in the field.
The updated paragraph is:
“In our opinion, the implementation of policies and strategies promoting the consumption of foods rich in provitamin A carotenoids could contribute to prevent the development of obesity and other metabolic diseases such as atherosclerosis and diabetes. Additionally, we also propose to introduce the quantification of retinoic acid and β-carotene into clinical practice as novel markers of health status. This is based on recent data obtained from clinical and preclinical models supporting the notion that low levels of these two compounds are tightly linked to the development of metabolic diseases”.
At the beginning the cell cultures studies were probably the most promising in terms of effects by β-carotene on reduced lipolysis. On the other hand, the poor transferability of the results (technical limitations) and the probably very high necessary and already toxic concentrations of retinoic acid (instead of β-carotene!) in the cell cultures experiments were pointed out.
Response: As the review acknowledges, the structure of this section starts with beta-carotene and continues with retinoic acid. It might be misleading, but we follow this structure throughout the entire manuscript.
We agree that retinoic acid doses used in many cell culture experiments are considered too high or even toxic, but the extensive use of these concentrations by notable researchers in the field cannot be ignored when writing a review. Also, we point out the concerns of this reviewer in the text, and we provide an example showing how retinoids do not efficiently cross plasma membranes on their free form on aqueous solutions such as cell culture media (Lines 244-246).
Concerning the animal studies, limitation regarding the β-carotene studies was also shown in particular. However, the authors own studies showed that too much retinyl ester in form of retinoic acid during adipose tissue development in rats could lead to obesity in later life [Ref 110]. Here, of course, I am missing a clear study regarding β-carotene, the actual topic of the review.
Response: Unfortunately, as the reviewer points out, there are not that many studies focused on beta-carotene and obesity. As part of my PhD studies back in Europe, we performed a study focused on this topic. This study is largely explained in section 4.4. on Animal Models of beta-carotene and obesity (citation 58,line 317). We thank the reviewer for pointing this out, as we did not add the corresponding citation to the explanation, which is now added to the end of the paragraph.
However, I would really advise the authors to start again from the beginning, and clearly work out the role of β-carotene in obesity research. To summarize and shorten the first parts in the introduction, cell- and animal studies with most relevant facts, in order to put the facts into relation with the relevant human studies on β-carotene and obesity research. I would especially work out the relevant human studies more thoroughly; especially here obvious mistakes in the citation had been also noticed.
Response: thank you for your suggestion. However we do not agree with the reviewer on starting this manuscript from the beginning. While we agree that human studies are very important, as mentioned before, we are convinced that cell culture and animal studies are crucial to understand the mechanism(s) of action of carotenoids and retinoids.
In addition, we agree that we have had a few obvious mistakes in the citations, and we take total blame for that. We have carefully revised all the citations throughout the manuscript and corrected them when necessary, including those mentioned by this reviewer.
After Line 88: Is table 1 really necessary??? If yes please provide conversion factors
Response: We consider that Table 1 is necessary. It provides basic, yet clear information of the amounts of the main provitamin A carotenoids and retinoids in our diet. According to the reviewer’s suggestion, we added the equivalence factors as a footnote.
In line 114: ‘carotenoid toxicity’ – I never heard about carotenoid toxicity; probably you mean vitamin A toxicity, and this is what you explain in the following sentences.
Response: Thank you for pointing out this mistake. We rephrased this statement to convey our statement as follows;
Lines 118-110: “The biggest advantage of this second approach is that β-carotene is present in the food matrix and absorbed in the intestine in a protein-mediated process”.
Comment, line 383-86: where is it proven that 9-cis or 13-cis beta-carotene have an adverse effect on human health? Are there studies (cell cultures, animals studies, human studies) showing effect by cis-isomers of beta-carotene? we also find cis isomer of beta-carotene in many food samples such as in fruits and green leafy vegetables.
Response: We apologize for any confusion. However, we do not state that 9-cis or 13-cis beta-carotene have any adverse effect on human health or any other model. This paragraph only indicates that beta-carotene is unstable when stored for a long time, exposed to elevated temperatures, light and/or oxygen. Our intention of showing Figure 2 is to indicate that different sources of commercially available β-carotene have different isomer composition. These different isomers could have different biological properties, either as a parent compound or when cleaved by the carotenoid-cleaving enzymes.
Thanks to the comment of this reviewer, we have rephrased this sentence to make it clearer to the reader, as it follows;
Lines 385-389: “Another possibility is that the purity of the β-carotene used on these trials was not the adequate, as β-carotene is unstable when stored for a long period, or when it is exposed to elevated temperatures, light, and/or oxygen [121-123].”
Last, Figure 2 shows that three different commercially available sources of β-carotene contain distinct racemic mixtures of this compound, which could result in differential biological properties.
Citations, references: I can't control all statements and literature cited; but the following are very general, imprecise and don't really reflect the details and results of the summarized studies.
Line108-110: ‘For examples, vitamin A fortification, of margarine in Denmark as well sugar fortification in Central America and Africa has considerably reduced the prevalence of vitamin A deficiency’[42-44].
First, what have it to do with beta-carotene? Did vitamin A fortification of margarine in Demark reduce vitamin A deficiency? I cannot find sugar fortification (with vitamin A?) in Central America (Brazil?) and Africa in the following 3 papers [42-44].
42. Keller, A., et al., A retrospective analysis of a societal experiment among the Danish population suggests that exposure to extra doses of vitamin A during fetal development may lower type 2 diabetes mellitus (T2DM) risk later in life. Br J Nutr, 2017. 117(5): p. 731-736.
Individuals from birth cohorts with the higher prenatal vitamin A exposure (born 1 December 1962–31 March 1964) and those with lower prenatal exposure (born 1 September 1959–31 December 1960) were followed up with regard to development of type 2 diabetes mellitus (T2DM) before 31 December 2012 in the Danish National Diabetes Registry and National Patient Register. Logistic regression revealed that higher fetal exposure (margarine vitamin A (RDA) concentration increased by 25%: retinol changed from 4.2 µg to 6 µg/g and b-carotene from 3.6 µg/g to 3 µg/g). Among the 193 803 included individuals, 101 178 and 92 625 were exposed and less-exposed, respectively, to the higher extra levels of vitamin A prenatally. The individuals exposed to higher vitamin A from fortification were less likely to develop T2DM (OR 0·88; 95 % CI 0·81, 0·95, P = 0·001) – But prevalence of vitamin A deficiency was not measured; the models only assessed risk of T2DM!
43. Palmer, A.C., et al., Provitamin A-biofortified maize increases serum beta-carotene, but not retinol, in marginally nourished children: a cluster-randomized trial in rural Zambia. Am J Clin Nutr, 2016. 104(1): p.181-90.
A study in Zambia assessed the impact of orange -maize meal (n=543 children) vs. white maize meal (n=481 children) among 4-8 years old children; 6-monthn biofortified maize intervention (higher daily intake of b-carotene – 133 µg daily RAE) revealed significant higher serum b-carotene (+0,14 µmol/L), but no difference in retinol concentrations between trial arms – no reduction in vitamin A deficiency and no sugar fortification!?!
44. Silva, L.L.S., et al., The impact of home fortification with multiple micronutrient powder on vitamin A status in young children: A multicenter pragmatic controlled trial in Brazil. Matern Child Nutr, 2017. 13(4).
A study in Brazil among 6-8 month old children, who received a multi-micronutrient powder (consisting of vitamin A, D, C, E, , B-vitamins (B1-B12) iron, zinc , copper, Iodine), who were followed for 4-6 month (n=399) and had significantly higher serum retinol, and therefore lower prevalence of vitamin A deficiency, but also lower prebalence of anemia and iron deficiency compared to 11-14 months old controls (n=395) - again, no sugar fortification!
Response: We apologize for the confusion with these references, and we thank the reviewer for pointing it out. We also appreciate the extensive time the reviewer took to look up and validate these references.
Following your comments, we decided to shorten this less relevant section by removing these two examples (lines 112-114), and adding a review discussing vitamin A supplementation strategies (lines 110-112). We believe that it will help the reader to focus on the main topic of the review, which is beta-carotene.
Line 405-410
‘Accordingly, Canas’ group showed that the supplementation of obese children with a carotenoid mixture showed a positive correlation between β-carotene concentration and decreased body mass index score, waist-to-height ratio, and adipose tissue size [126]. Similarly, supplementation of β-cryptoxanthin, another provitamin A carotenoid, to obese Japanese women resulted in increased serum levels of β-cryptoxanthin, reduction in body weight, visceral fat and waist circumference [127, 128]’.
Response: We thank the reviewer for his/her careful explanations and summary of these three papers. Thanks to his/her input, we rephrased these citations, and carefully edited our statements. While we agree with the reviewer’s comments that, in some cases, the sample size could seem small, the goal of this review was not to undermine the statistical validity of specific publications. As the reviewer knows, collecting human data can sometimes be limiting, and the statistical validity of the studies should be tested prior performing the studies by using a power analysis measurements. Therefore, when we decided to cite these papers assuming the authors performed these studies the best they could.
126. Canas, J.A., et al., Effects of Mixed Carotenoids on Adipokines and Abdominal Adiposity in Children: A Pilot Study. J Clin Endocrinol Metab, 2017. 102(6): p. 1983-1990.
A very small pilot study (n= 17 children, 10 years old) assessing mixed carotenoid supplementation (MCS) versus placebo on adipokines and the accrual of abdominal adiposity in children with simple obesity. After 6 month the children who received carotenoid mixture (n=8) had a significantly higher alpha- and beta-carotene, lutein and zeaxanthin, but significantly reduction in serum retinol; BMI z-scores increased by 4% in the placebo group but decreased by 4% in the MCS group; further high-molecular-weight adiponectin (HMW-ADI) increased by 79% in the MCS vs. placebo group. – the authors suggested a putative beneficial role of beta-carotene in the prevention and management of obesity.
Response: For this first statement, we have now rephrased our manuscript and added;
“Accordingly, Canas’ group showed that the supplementation of obese children with a carotenoid mixture showed a significant increase in plasma concentration of various carotenoids including β-carotene, accompanied by a decreased body mass index score, waist-to-height ratio, and subcutaneous adipose tissue content [127].”
127. Hirose, A., et al., Higher intake of cryptoxanthin is related to low body mass index and body fat in Japanese middle-aged women. Maturitas, 2017. 96: p. 89-94.
Different intake pattern regarding higher intake of cryptoxanthin (especially high in mandarins) was associated with low BMI (< 22 kg/m2 (n=56) vs >22 kg/m2 (n=23)) and low body fat percentage (<25%, n=35 vs >25%, n=53).
Limitation of the study: small number (n=88 women), data were taken from a baseline survey; most women were not obese, and plasma cryptoxanthin were not measured – this was not a supplementation study, and cryptoxanthin was not supplemented to obese women as written in line 408!
128. Iwamoto, M., et al., Supplementation of highly concentrated beta-cryptoxanthin in a satsuma mandarin beverage improves adipocytokine profiles in obese Japanese women. Lipids Health Dis, 2012. 11: p. 52.
An intervention study in which β-cryptoxanthin (4.7 mg/day) was given for 3 weeks to 17 moderately obese postmenopausal women. The result indicated no changes in body weight! serum cryptoxanthin increased by 4-fiold, and high molecular weight-adiponectin significantly. The authors concluded that increasing the intake of β-cryptoxanthin to approximately 4 mg per day for 3 weeks may have beneficial effects on the serum adipocytokine status and consequently alleviate progression of metabolic syndrome.
Supplementation of β-cryptoxanthin to moderate obese (BMI=27 kg/m2) Japanese women DID NOT result in reduction of body weight, visceral fat and waist circumference, as mentioned in Line 409-10!! Visceral fat and waist circumference were not measured at all!
Response: For the second statement, we apologize for the confusion. As the reviewer states above, the first study (citation 127) was not performed in obese women, and only citation 128 was referred to obese women. We decided to merge both citations in one comprehensive sentence, as the main message of both studies were very similar; beta-cryptoxanthin reduces obesity.
As we focus mostly on the effect of beta-carotene in obesity, and following this reviewer’s suggestions such as the low number of subjects for citation 127, together with the lack of measurements related to obesity endpoint parameters for citation 128, we decided to remove this sentence from the manuscript.
I cannot control all the 149 citations, and I really hope that most citations reflect the truth better than the above-mentioned lines and references.
Response: We understand that the reviewer can be concerned about the way we cited certain publications. However, we are confident he/she will also appreciate and understand the responses we provided, and that these inaccuracies did not have a malicious intent. Additionally, we have revised the accuracy of the citations in the paper. To the best of our knowledge, no other errors were found.
Reviewer 3 Report
In this review article, Coronel et al. summarized current
knowledge on provitamin A carotenoids as the preferred source of vitamin A for
humans and highlighted what is currently known about the role of β-carotene and
vitamin A in obesity research. The authors covered the main findings related to
β-carotene and its derivatives (retinoids) on energy metabolism, adipocyte
function, and adipose tissue biology in various experimental models and
discussed the technical limitation occurred in these model system. The review
is well organized and supported by the references.
Author Response
Reviewer #3
In this review article, Coronel et al. summarized current knowledge on provitamin A carotenoids as the preferred source of vitamin A for humans and highlighted what is currently known about the role of β-carotene and vitamin A in obesity research. The authors covered the main findings related to β-carotene and its derivatives (retinoids) on energy metabolism, adipocyte function, and adipose tissue biology in various experimental models and discussed the technical limitation occurred in these model system. The review is well organized and supported by the references.
Response: We want to thank you for your positive comments on our work.
Round 2
Reviewer 2 Report
Table 1; Ref 33: equivalence factors as a footnote, cite properly:
Food and Nutrition Board, Institute of Medicine. (2001). Dietary reference intakes for vitamin A, vitamin K, arsenic, boron, chromium, copper, iodine, iron, manganese, molybdenum, nickel, silicon, vanadium, and zinc. National Academy of Medicine.
Author Response
Thank you for your suggestion. We have modified the reference accordingly (marked in yellow).
